# Genetic Screening of Targeted Region on the Chromosome 22q11.2 in Patients with Microtia and Congenital Heart Defect

**DOI:** 10.3390/genes14040879

**Published:** 2023-04-07

**Authors:** Caiyun Zhu, Yang Yang, Bo Pan, Hui Wei, Jiahang Ju, Nuo Si, Qi Xu

**Affiliations:** 1State Key Laboratory of Medical Molecular Biology, Institute of Basic Medical Sciences, Chinese Academy of Medical Sciences, School of Basic Medicine Peking Union Medical College, Beijing 100005, China; 2Neuroscience Center, Chinese Academy of Medical Sciences, Beijing 100005, China; 3Department of Auricular Reconstruction, Plastic Surgery Hospital, Chinese Academy of Medical Science and Peking Union Medical College, Beijing 100144, China; 4Research Center, Plastic Surgery Hospital, Chinese Academy of Medical Science and Peking Union Medical College, Beijing 100144, China

**Keywords:** microtia, congenital heart defect, target capture sequencing, *TBX1*, *CLTCL1*

## Abstract

Microtia is a congenital malformation characterized by a small, abnormally shaped auricle (pinna) ranging in severity. Congenital heart defect (CHD) is one of the comorbid anomalies with microtia. However, the genetic basis of the co-existence of microtia and CHD remains unclear. Copy number variations (CNVs) of 22q11.2 contribute significantly to microtia and CHD, respectively, thus suggesting a possible shared genetic cause embedded in this genomic region. In this study, 19 sporadic patients with microtia and CHD, as well as a nuclear family, were enrolled for genetic screening of single nucleotide variations (SNVs) and CNVs in 22q11.2 by target capture sequencing. We detected a total of 105 potential deleterious variations, which were enriched in ear- or heart-development-related genes, including *TBX1* and *DGCR8*. The gene burden analysis also suggested that these genes carry more deleterious mutations in the patients, as well as several other genes associated with cardiac development, such as *CLTCL1*. Additionally, a microduplication harboring *SUSD2* was validated in an independent cohort. This study provides new insights into the underlying mechanisms for the comorbidity of microtia and CHD focusing on chromosome 22q11.2, and suggests that a combination of genetic variations, including SNVs and CNVs, may play a crucial role instead of single gene mutation.

## 1. Introduction

Microtia is defined as a developmental malformation of the external ear, characterized by a small, abnormally shaped auricle (pinna), unilaterally or bilaterally, that ranges from mild structural abnormalities to complete absent ear. The estimated incidence rate of microtia ranges from 0.8 to 17.4 per 10,000 births [1,2,3]. Microtia may manifest as an isolated condition or as part of a spectrum of abnormalities or a syndrome, such as Velo-Cardio-Facial syndrome, CHARGE syndrome, and Goldenhar syndrome [4,5,6]. A total of 85% of patients with CHARGE syndrome also had congenital heart defect (CHD). A recent study in the Chinese Han population showed that 44% of patients with microtia have one or more comorbid anomalies, including facial and neck (40%), followed by the musculoskeletal system (35%) and cardiovascular system (11%) [7]. A high incidence of CHD (3.35~18.5%) in microtia patients has been observed in multiple populations, but the potential genetic mechanisms underlying this association are still unclear [8].

The previous genetic studies in microtia have focused on the structural variation (SV) in the chromosome 22q11.2 region. The 22q11.2 region is a hotspot for chromosomal rearrangements mediated by non-allelic homologous recombination (NAHR) between eight low-copy repeats (LCRs), known as LCR22-A to LCR22-H, which share 97–98% identity [9,10]. The abnormal rearrangements (including duplications, deletions, and translocations) of this region are associated with a variety of diseases, including microtia and CHD. Microdeletions and microduplications are the most frequent SVs found in 22q11.2. Previous studies showed that 22q11.2 deletion syndrome (22q11.2 DS), in which most patients share a ~3 Mb heterozygous deletion between LCR22-A and LCR22-D, is the most common cause of syndromic palatal anomalies and the second-most common cause of CHD and developmental delays [11,12,13,14]. The physiological phenotypes in patients with 22q11.2 microduplications are extremely variable, ranging from multiple defects to mild learning difficulties. However, they share features including a heart defect, urogenital abnormalities, and velopharyngeal insufficiency. Among the various phenotypes, both ear malformations and heart malformations are represented in 22q11.2-related syndromes, respectively [15,16].

Considering that ear and cardiac malformations are frequently present in the phenotype spectrum of complex diseases associated with the 22q11.2 region, we hypothesize that pathogenic genetic variants residing in this region may contribute to both ear and heart dysplasia. In this study, the entire 8 Mb genomic region of 22q11.2 was screened by a target capture deep sequencing strategy in patients with both microtia and CHD, to identify potential pathological SNVs and CNVs, and analyze mutational burden in 22q11.2.

## 2. Materials and Methods

### 2.1. Participants

A total of 19 sporadic patients, diagnosed with microtia and CHD, and one nuclear family were enrolled as discovery set in this study. In total, 96 healthy controls and 64 patients with isolated microtia or microtia and other abnormalities were included for validation. All patients were unrelated and diagnosed by clinical evaluation in Plastic Surgery Hospital (Institute), Chinese Academy of Medical Sciences (CAMS), Peking Union Medical College (PUMC). In addition to microtia and CHD, the child in the nuclear family was diagnosed with epilepsy and polydactyly. The severity of microtia was ascertained using the classification system proposed by Hunter [17]. First-degree microtia is defined as the presence of all the normal ear components and the median longitudinal length more than 2 SD below the mean. Second degree microtia is characterized by a median longitudinal length of the ear more than 2 SD below the mean in the presence of some, but not all, parts of the normal ear. Third degree microtia is defined as the presence of some auricular structures, but none of these structures conform to recognized ear components. Each patient from the cohort underwent a detailed physical examination by the clinical authors.

The study was approved by the Ethics Committee of the Chinese Academy of Medical Sciences and Peking Union Medical College. Informed written consent was obtained from all participants and their guardians.

### 2.2. Targeted Region Capture Sequencing

Genomic DNA from the peripheral blood of participants was extracted using the QuickGene DNA Whole Blood Kit L (KURABO) and quantitated by Qubit^®^ 2.0 Fluorometer (Life Technologies, Grand Island, New York, NY, USA). The custom-targeted sequence capture probes were used to capture the 22q11.2 region (8M, including coding and non-coding regions). The construction of the Illumina sequencing libraries was performed using the SureSelectXT Custom 12–24 Mb kit in accordance with the manufacturer’s instructions. After quality control, the libraries were sequenced in the Illumina HiSeq 2500 platform to generate 150 bp paired-end reads.

### 2.3. Quality Control of Sequencing Data and Single Nucleotide Variant (SNV) Calling

The quality control of raw data was performed with FASTQC (version 0.11.5) [18]. Known Illumina primers and adapter sequences were removed from each read with cutadapt (version 1.18) [19]. Bases lower than a defined Phred quality score (default: 20) at the 3′ end were trimmed off from each read. Reads with more than 30% of low-quality nucleotides (≤20) and reads with more than 5% of non-determinant nucleotides (N) were filtered out using FASTX_Toolkit software. The raw reads that passed the quality control were considered clean reads and aligned to the NCBI human reference genome (hg19) using Burrows–Wheeler Aligner (BWA version 0.7.17-r1188) [20]. Picard tools (version 2.18) (https://broadinstitute.github.io/picard/, accessed on 15 March 2019) and SAMtools (version 1.8) were used to sort, mark duplicate reads and reorder the bam alignment results of each sample [21]. We then performed local realignment and recalibration of the BWA-aligned reads using the Genome Analysis Toolkit (GATK version 3.7) [22]. GATK’s HaplotypeCaller was used to call SNV across all samples simultaneously in the multi-sample calling mode, which allowed us to distinguish homozygous reference sites and sites with missing data. To obtain a reliable list of variants suitable for further analysis, single nucleotide polymorphism (SNP) and insertion/deletion (indel) were filtered separately. SNPs were filtered by the following parameters: QD < 2.0, FS > 60.0, SOR > 4.0, MQ < 40.0, MQRankSum < −12.5, ReadPosRankSum < −8.0. Indels were filtered by the following parameters: QD < 2.0, FS > 200.0, SOR > 10.0, InbreedingCoeff < −0.8, ReadPosRankSum < −20.0. Minor allele frequency (MAF) of each SNP in the patient population was computed using VCFtools (version 0.1.15) [23].

### 2.4. Functional Annotation and Gene Burden Analysis

Variants were considered novel if not annotated in dbSNP (release 147) and the 1000 Genomes Project (phase 3). Rare variants were defined as not exceeding 1% allele frequency based on the East Asian population in the 1000 Genomes Project (phase 3) and the ExAC (Exome Aggregation Consortium) database [24,25]. Novel and rare variants were further categorized as loss-of-function (LoF) (i.e., stop-gain, frameshift, or splice site alterations) and missense damaging (missense predicted damaging by at least two out of eight databases). Eight databases were used to predict the effect of amino acid changes on protein function, including SIFT, Polyphen2, LRT, MutationTaster, Mutation Assessor, FATHMM, GERP++, and CADD [26]. The SNVs that were predicted to be deleterious in two out of the eight databases were selected as potential candidates. Gene-based testing was performed using PLINK/SEQ (https://zzz.bwh.harvard.edu/plinkseq/, accessed on 24 July 2019) one-sided burden test or SKAT-O with Han Chinese in Beijing, China (CHB) in the 1000 Genomes Project (phase3) served as controls. The significance of the gene burden results is evaluated with 10,000 permutations. Potential deleterious variants were further annotated with the SCAN database to assess their effects on expression levels of known causal genes [27]. The Gene Ontology (GO) enrichment analysis was performed using the clusterProfiler package, setting a q-value threshold of 0.05 for statistical significance [28].

### 2.5. Copy Number Variant (CNV) Analysis and Validation by qRT-PCR

CNV analysis from the targeted sequencing data were performed using CNVnator with a bin size of 100 bp (version 0.3.3) [29]. CNV calls with a *p*-value < 0.01 and a size > 1 kb remained, and calls with >50% of q0 (zero mapping quality) reads within the CNV regions were removed. Two calls were considered concordant if they had >50% reciprocal overlap. Concordant calls were merged by selecting the largest coordinates of each bound, pursuing all concordant calls were covered by merged CNV. CNVs with a recurrence rate > 60% of patients and not present in the parents of the nuclear family were retained for further validation. CNVs were annotated with the Reference Sequence (RefSeq) database and the Database of Genomic Variants (DGV, http://dgv.tcag.ca, accessed on 16 February 2020) with BEDTools using −r 0.5 option (i.e., 50% overlap) [30,31]. A CNV was classified as rare if ≤50% of its length overlapped regions present at >1% frequency in DGV.

Six candidate CNVs were confirmed by quantitative real-time PCR (qRT-PCR) using primers specific to the region (Appendix A). Each sample was tested three times and the ∆∆Ct method was used to calculate the relative copy number (CN) for each region. The linear regression analysis adjusted for age and sex was performed to assess the difference in copy number between patients and controls in the validation set. The sequences of the primers and probes used in this study are shown in Appendix A.

## 3. Results

### 3.1. Clinical Description

We included 19 sporadic patients (mean age 12 years), 5 (26.32%) females and 14 (73.68%) males, with a diagnosis of microtia and CHD in the discovery set. The major clinical features of these patients are shown in Table 1. Most patients (57.89%) had second-degree microtia and six patients (31.58%) had first-degree microtia. The two patients with the most serve form of microtia (third-degree) also had craniofacial anomalies, such as facial cleft, cleft palate, and hemifacial dysplasia. The most common cardiac abnormality in the discovery set was ventricular septal defect (VSD) (observed in nine patients), followed by an atrial septal defect (ASD) (21.05%) and tetralogy of fallot (TOF) (21.05%). The mean ages for patients and controls in the validation set were 11.39 and 39.23 years, respectively. Males accounted for 57.81% and 40.63% of the patients and controls, respectively.

### 3.2. Functional Annotation of SNVs

In this study, about 5.04 Gb of raw sequencing data were generated for each sample. After stringent quality assessment and data filtering, the samples yielded approximately 28.6 million (85.18%) high-quality reads each, with depths ranging from 65× to 94× (Appendix A). Based on the read depth smoothed in a 100 bp sliding window, a nearly 3 Mb deletion across LCR22-AD was observed in the child of the nuclear family (Figure 1).

After quality control, we included a total of 37,659 SNVs (29,071 SNPs and 8584 indels) in 19 sporadic cases. Figure 2a shows that 21.34% of SNPs were observed in only one case, while 7.53% (2188) of SNPs were observed in all 19 cases. There was a total of 396 SNPs distributed in the coding regions, including 203 non-synonymous sites, 184 synonymous sites, five stop-gain sites, and two unknown sites, in addition to two splice sites (Figure 2b). Figure 2c shows that the length of most indels (92.70%) was less than 15 bp. Functional annotation indicated there were ten LoF indels, including nine frameshift indels and one stop-gain indel (Figure 2d).

### 3.3. Identification of Genes with Potential Functional Consequences

The primary objective of the study was to detect variants with potential functional consequences associated with microtia and CHD. We identified 105 LoFs or functional missense variants from 2165 novel and 82 rare variants from 19 sporadic cases. GO enrichment analysis illustrated that genes harboring the 105 candidate SNPs were enriched in biological processes such as the cellular modified amino acid catabolic process, the leukotriene D4 metabolic process, and the glutathione catabolic process, and molecular function such as omega peptidase activity and transferase activity (Figure 3a). Of these putatively deleterious variants, 50 variants were absent in the healthy parents and resided in 38 genes, such as *TBX1* and *DGCR8*. *TBX1* harboring rs5993826 and a novel SNP, and *DGCR8* harboring rs182736423 and a novel SNP were known to be associated with the development of the pharyngeal arches or CHD. Figure 3b shows that each sample harbored one to six variants of the 50 variants. Additionally, four SNPs (rs368014, rs11704009, rs9623076, rs4820360) were predicted to be eQTLs for the expression of *CXCL10* (*p* < 0.0001) (Appendix A). rs738802 and rs396330 could influence the expression levels of *DGCR2*, *DGCR11*, and *DGCR12* (*p* < 0.0001).

A total of 74 overlapped rare and potential deleterious variations were obtained between 19 cases and 103 controls (CHB derived from the 1000 Genome Project). Gene-based testing between cases and CHB illustrated that these potential causal variants were significantly enriched in 10 genes (Table 2). The top gene on the list—*SUSD2*—had a nearly 22-fold higher frequency in patients than in controls. We also assessed the genetic burden based on candidate genes associated with microtia and CHD. The potential deleterious variants were also significantly enriched in *TBX1* (the classical 22q11.2 DS phenotype associated gene) and *CLTCL1* (the classical CHD associated gene). Additionally, 20 genes harbored potential deleterious variations that only appeared in the sporadic patients, including *RIMBP3* (harbored 17 SNPs), *DGCR8*, *PI4KA*, and *UFD1*.

### 3.4. Microduplication Fragment May Associate with Microtia and CHD

The sequencing depth for each sample varied from 65× to nearly 95×, allowing sufficient power to detect CNV. For 19 sporadic cases and healthy parents, a total of unique 1135 CNVs were identified with an average of 54 calls per sample (Appendix A). Calls across all samples affected nearly 28.81% (2.31 Mbp) of the targeted region. The CNV calls had an average length of 14.56 kbp, with a median length of 9.1 kbp (Figure 4a). The cohort showed more duplications than deletions (*p* < 0.001), where duplications had a median length of 12.95 kbp compared to deletions at 3.90 kbp. The number of sample-specific CNVs for each sample ranged from 5 to 34.

We identified recurrent CNVs between samples as calls with at least 50% reciprocal overlap. After removing merged calls present in the healthy parents, six recurrent CNV calls were retained as candidate CNVs, including four duplications and two deletions (Figure 4b). Except for DUP4, all candidate CNV calls were rare CNVs and not overlapped with segmental duplications (SDs). In our study, 11 of 19 cases had CNV amplifications (DUP3) overlapping most of the sushi domain containing 2 (*SUSD2*), which was validated by qPCR in the validation set (*p* < 0.001, Figure 4b). The results of the analysis are summarized in Appendix A.

## 4. Discussion

Our study shows that the clinical manifestations of microtia and CHD comorbidity are highly heterogeneous. Consistent with previous studies, we found that microtia occurs more frequently in males and on the right side (unilateral) [32,33,34]. Nearly 60% of the patients were classified as having second-degree microtia. CHD is the most common congenital disorder in newborns. A previous study showed that Asia has the highest CHD birth prevalence, with 9.3 per 1000 newborns [35]. Among patients with microtia, ASD is the most common lesion, followed by VSD and PDA [8]. In our study, the top three kinds of CHD were VSD, ASD, and TOF.

Considering the high co-occurrence of these two complex diseases, it is speculated that common genetic mutations or biological pathways may play important roles in both microtia and CHD. Among the syndromes with these symptoms, Goldenhar syndrome is also associated with this region in addition to those directly caused by the 22q11.2 variations. Balcı et al. suggested that each patient with Goldenhar syndrome should be screened for 22q11.2 deletion, which could be a candidate causal gene for this syndrome [36]. The strategy of capture-based target enrichment followed by deep sequencing (~80×) enabled us to identify both SNVs and small CNVs simultaneously. Only the child in the nuclear family had a 3 Mbp fragment deletion (LCR22-A to LCR22-D), which belongs to the typical 22q11.2 DS and contributes to the complex phenotypes of the patient [11].

Among 19 sporadic cases, LoF variants and putatively functional missense SNVs were combined for a total of 105 predicted deleterious variants. These variants are predicted to have a strong effect on protein structure and function. Each patient carries at least one harmful mutation, especially three individuals who carry variants of known susceptible genes for microtia or CHD. Two patients (4,12) had a different *TBX1* deleterious variant, and one patient (5) had two deleterious variants in *DGCR8*.

*TBX1* is a member of the phylogenetically conserved T-box transcription factor family. *TBX1* is highlighted as the most widely studied gene in 22q11.2 DS and is required for the normal development of the fourth pharyngeal arch arteries [37,38]. The previous study suggested that *TBX1* mutation is responsible for five major phenotypes in 22q11.2 DS, including abnormal facies, cardiac defects, thymic hypoplasia, velopharyngeal insufficiency with cleft palate, and parathyroid dysfunction with hypocalcemia [39]. Although *TBX1* is not expressed in NCCs, NCC patterning of both the surface ectoderm and the second heart field is affected in conditional mutants [40,41]. Haploinsufficiency of *TBX1* leads to partially penetrant cardiovascular, thymic, and parathyroid defects in mice [42,43]. Another gene of interest is *DGCR8*, encoding an essential component of the microprocessor complex that mediates the biogenesis of microRNA [44]. The abnormal microRNA biogenesis due to *DGCR8* deficiency contributes to structural abnormalities in the heart and vasculature and neuronal deficits [45,46]. In mice, loss-of-function of *DGCR8* in NCCs results in congenital heart abnormalities that are characteristic of 22q11.2 DS, and heterozygous loss-of-function mutations of *DGCR8* result in neuronal deficits [45,47].

Gene-based enrichment analysis showed that these variations were significantly enriched in 10 genes, such as *SUSD2*, *TBX1*, *CLTCL1*, and *TANGO2*. The ratio of deleterious mutation burden between cases and controls reached 21 in gene *SUSD2*. As well as a tumor-reversing protein, *SUSD2* (encoding sushi domain containing 2) also plays a dual role during neural development [48,49,50]. *SUSD2* can alter axon and dendritic outgrowth in young neurons, and regulate excitatory synapse numbers at a later stage. *CLTCL1* encodes a major protein of the polyhedral coat of coated pits and vesicles, which is associated with multiple syndromes, including DiGeorge syndrome and Velo-Cardio-Facial syndrome. Based on whole exome sequencing and the analysis of the rare mutations burden, Shi et al. found that *CLTCL1* was associated with the total anomalous pulmonary venous connection (TAPVC), and highly expressed in human umbilical vein endothelial cells and human embryonic hearts [51]. *TANGO2* (transport and Golgi organization 2 homolog) encodes a protein that is involved in redistributing Golgi membranes into the endoplasmic reticulum [52]. Initial reported *TANGO2*-related disorders were considered metabolic encephalopathy and arrhythmias, characterized by variable clinical features [53,54]. The variability in presentations was expanded to primarily neurological presentations, such as progressive neurodegeneration and cognitive impairment [55]. Further study is needed to assess the contribution of *TANGO2* to microtia and CHD, which is involved in the 22q11.2 phenotypic spectrum. In accordance with the previous study, genes harbored by these identified SNVs can only explain a fraction of the genotype–phenotype relationships underlying microtia and CHD [56].

Additionally, potential pathogenic variations that appear only in patients are mainly enriched in some genes, such as *RIMBP3*, *DGCR8*, *PI4KA*, and *UFD1*. *RIMBP3* is mostly expressed outside of the brain. As a member of the Rab3-interacting molecules (RIM), binding proteins (RIMBPs), RIMBP3, together with RIMBP1 and RIMBP2, regulate the fidelity of synaptic transmission at central synapses and play a potential role in regulating the fusion and release of neuronal vesicles by stabilizing the active zone structure in the presynaptic region and locating presynaptic ion channels [57,58]. Previous studies have suggested that *PI4KA* is a candidate gene for schizophrenia susceptibility and plays an important role in early brain development [59,60]. Ubiquitin Fusion-Degradation Protein-1 Like (Ufd1l) encoded by the gene *UFD1* serves as a ubiquitin recognition factor linking the ubiquitination and degradation process of proteins. p97 in mammals associated with the cofactor Ufd1-Npl4 forms a functional conservative complex, which can mediate the transport of endoplasmic reticulum proteins into the cytosol for degradation [61,62,63]. As a housekeeping gene, *UFD1* is mainly expressed in pharyngeal arches, palatal precursors, and ears, heart, and brain during development, and the haploinsufficiency of *UFD1* is supposed to be responsible for the pathogenesis of 22q11.2 DS [64,65,66]. A previous study reported the association between the bicuspid aortic valve and the reduced expression of *UFD1*, further supporting the role of *UFD1* in normal heart formation [67].

The CNV of genomic fragments is a common phenomenon that affects about 4.8–9.5% of the human genome [68]. The enrichment of CNVs in SDs was consistent with results from the previous study [69]. The hotspots of chromosomal rearrangement defined by SDs can act as mediators of normal variation as well as genomic diseases [68]. Previous studies combined CNVs and SNPs and illustrated the association of CNVs with multiple diseases, such as autism, neuroblastoma, and severe early-onset obesity [70,71,72]. Among six CNVs after strict filtering, a novel microduplication that harbored gene *SUSD2* was identified as being associated with phenotypes of patients, and confirmed by quantitative PCR. *SUSD2* also plays a dual role at different stages of neural development [50]. However, the molecular mechanism underlying microtia or CHD and *SUSD2* remains unknown, and further studies will be required to address this issue.

Given the high heterogeneity of microtia and CHD, the comorbidity of these two diseases is unlikely to be caused by a few specific loci but rather may be the result of the combined effects of multiple variations in multiple genes. In our study, deleterious variants were significantly enriched in serval genes (*SUSD2*, *TBX1*, *DGCR8*, *CLTCL1*, *TANGO2* and *UFD1*) and the microduplication fragment that may present genetic factors relevant to the etiology. The candidate variants and genes deserve further exploration to better understand the specific mechanisms of microtia and CHD comorbidity. The small sample size is the main limitation of our study. To increase the statistical power of the gene-based rare variant association method, it is usually necessary to increase the sample size [73]. An in-depth study of screening for the whole genome may provide new insights for the investigation of rare and complex diseases.

## Figures and Tables

**Figure 1 genes-14-00879-f001:**
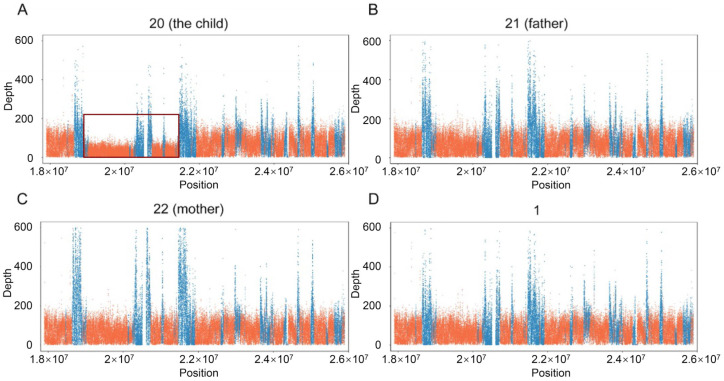
Depth of several samples with a 100 bp sliding window. (**A**) The child in the nuclear family. The red box indicates the deletion of LCRA-D. (**B**,**C**) The healthy parents in the nuclear family. (**D**) Patient #1. Bases labeled as N in the reference genome are represented by blue dots, while red dots indicate bases located in other regions.

**Figure 2 genes-14-00879-f002:**
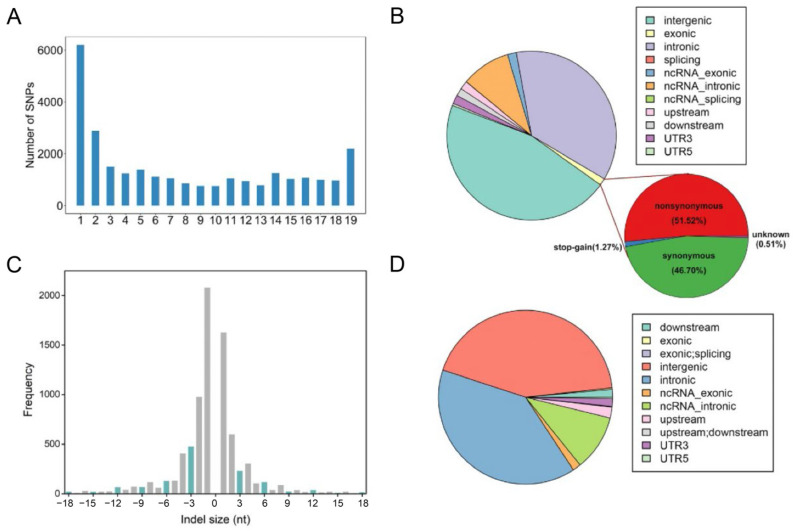
Distribution of SNVs by location and type. (**A**) Frequency distributions in terms of occurrences in 19 cases. (**B**) Distribution of SNPs by their gene location and type. (**C**) Distribution of indel lengths. (**D**) Distribution of indels by their gene location.

**Figure 3 genes-14-00879-f003:**
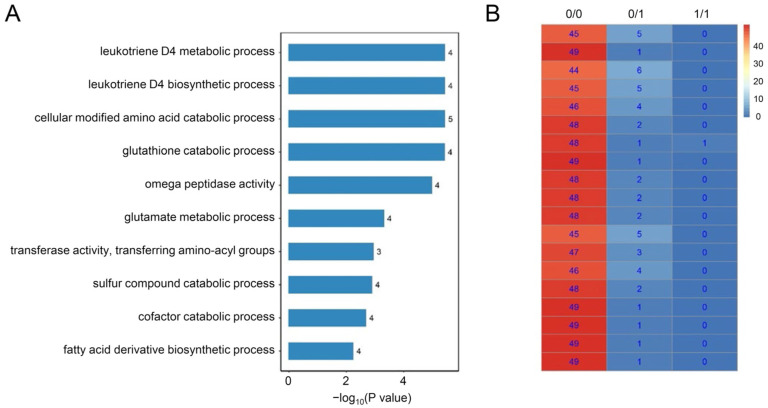
(**A**) GO enrichment analysis of genes harbored 105 LoFs and missense SNPs. (**B**) Distribution of candidate SNPs (SNP-1) in 19 cases. 0/0, homozygote for the reference allele; 0/1, heterozygote; 1/1, homozygote for the alternative allele.

**Figure 4 genes-14-00879-f004:**
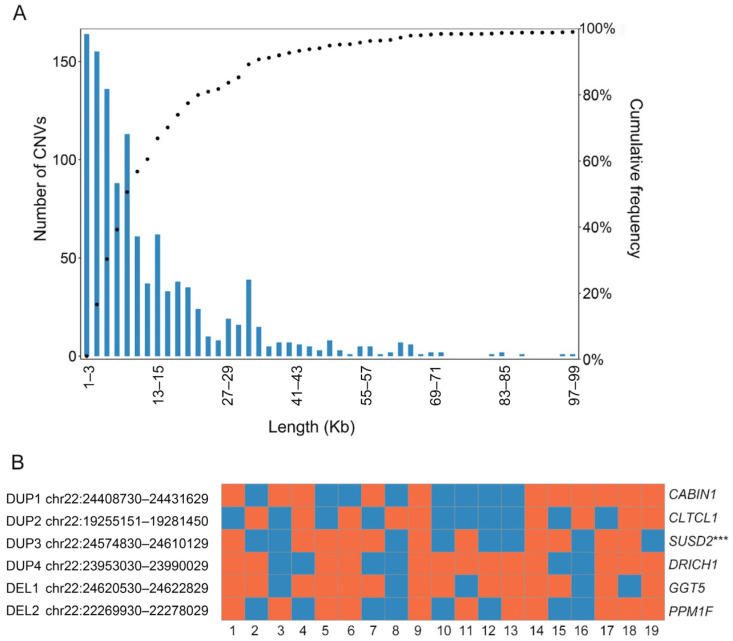
CNVs called by CNVnator. (**A**) Length distribution of CNV calls across 19 cases and the healthy parents. (**B**) Distribution of candidate CNVs in 19 cases. The presence of a particular copy number variation (CNV) in the patient is indicated by red, and its absence is indicated by blue. *** *p* < 0.001 in qRT-PCR.

**Table 1 genes-14-00879-t001:** Summary of the clinical features observed in 20 patients included in our study.

Patient	Sex	Age	Ear Anomalies	Cardiac Anomalies	Other Anomalies
1	M	8	B; degree I	VSD	
2	M	6	R; degree I	VSD	
3	F	20	R; degree I	TOF	
4	M	13	L; degree I	VSD; TI; PI	
5	M	25	L; degree I	VSD	
6	M	9	L; degree I	TOF	
7	M	6	R; degree II	PDA	
8	M	12	R; degree II	VSD; PH	
9	M	11	R; degree II	VSD	cleft palate
10	M	8	R; degree II	TOF	
11	M	7	R; degree II	PFO	bilateral preauricular fistula
12	F	13	R; degree II	ASD	
13	M	10	R; degree II	VSD	
14	M	18	L; degree II	VSD	
15	M	15	L; degree II	ASD	
16	F	6	L; degree II	VSD	
17	M	7	L; degree II	TOF	
18	F	35	R; degree III	ASD	facial cleft; hemifacial dysplasia
19	F	8	R; degree III	ASD	hemifacial dysplasia; facial transversal cleft; left eye cyst
20 ^1^	M	7	B; degree I	ASD; PDA	polydactyly; epilepsy

^1^ The child in the nuclear family. F, feminine; M, masculine; L, left; R, right; B, bilateral; VSD, ventricular septal defect; ASD, atrial septal defect; TOF, tetralogy of fallot; TI, tricuspid insufficiency; PDA, patent ductus arteriosus; PFO, patent foramen ovale; PH, pulmonary hypertension; PI, pulmonary insufficiency. Blank boxes indicate the absence of the phenotype. Degree I, first degree microtia; degree II, second degree microtia; degree III, third degree microtia.

**Table 2 genes-14-00879-t002:** Gene-based association between cases and CHB.

Gene	Counts (CHB) ^1^	Counts (Cases)	Frequency Ratio	*p* Value
*SUSD2*	5	20	21.68	<0.0001
*COMT*	9	2	1.20	<0.0001
*GGT2*	16	22	7.45	<0.0001
*TOP3B*	16	4	1.36	<0.0001
*SCARF2*	29	6	1.12	<0.0001
*PIWIL3*	1	3	16.26	0.0001
*BCR*	2	2	5.42	0.0002
*TBX1*	38	8	1.14	0.0293
*TANGO2*	1	1	5.42	0.0350
*CLTCL1*	116	25	1.17	0.0430

^1^ Counts represent the number of times that mutation is observed in cases or CHB.

## Data Availability

All the raw sequence data have been deposited in the National Genomics Data Center (NGDC) Genome Sequence Archive (GSA) under the accession number: HRA000684 (https://ngdc.cncb.ac.cn/gsa-human/s/mxTu6mt8, accessed on 5 January 2023).

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
