# Peer review of "Genetic Screening of Targeted Region on the Chromosome 22q11.2 in Patients with Microtia and Congenital Heart Defect"

_genes, 2023, doi:10.3390/genes14040879_

Round 1

Reviewer 1 Report

The paper seems to report the results of an interesting study. To my opinion Discussion could be clearer, perhaps a Table summarizing the data would be useful, conclusions should be made.

Author Response

Point 1: The paper seems to report the results of an interesting study. To my opinion Discussion could be clearer, perhaps a Table summarizing the data would be useful, conclusions should be made.

Response 1: Thank you for your suggestions. To make the conclusion more clearer and explicit, we have added a supplementary table to summarize all the results and reorganized the discussion section (Table S5) (lines 242-243, 250-252, 258-265, 270-272, 310-327, 339-345). In addition, we have carefully considered your comments regarding the need for a moderate revision of the English language and have made the necessary changes to improve the readability and clarity of the manuscript. We appreciate your valuable suggestions, which have helped us improve our work's quality.

Please refer to Table S5 in the attachment.

Reviewer 2 Report

The paper explores the genetic basis for the comorbidity of microtia and congenital heart defects (CHD), which are both associated with a series of genetic craniofacial syndromes. The study uses target capture sequencing to scan for potential pathological single nucleotide variations (SNVs) and copy number variations (CNVs) in the 22q11.2 region of the genome in 19 sporadic patients with microtia and CHD, as well as a nuclear family. The authors detect a total of 105 potential deleterious variations, which are enriched in ear or heart development-related genes. The paper also discusses genes and functions potentially affected by these deleterious variations. The study suggests that a combination of genetic variations, including SNVs and CNVs, may play a crucial role in the comorbidity of microtia and CHD. The paper provides new insight into the underlying mechanisms for the comorbidity of microtia and CHD and highlights the importance of genetic analysis in understanding the genetic basis of complex disorders. Overall, the study is well-structured and provides clear and concise methods, making it a valuable contribution to the field of microtia and CHD genetics.

1.     The paper presents a clear and concise description of the methods and results for SNVs and CNVs reports. The discussion section may benefit from a minor revision, as the first paragraph appears somewhat repetitive when compared to the first paragraph of the introduction.

Author Response

(x) English language and style are fine/minor spell check required

The paper explores the genetic basis for the comorbidity of microtia and congenital heart defects (CHD), which are both associated with a series of genetic craniofacial syndromes. The study uses target capture sequencing to scan for potential pathological single nucleotide variations (SNVs) and copy number variations (CNVs) in the 22q11.2 region of the genome in 19 sporadic patients with microtia and CHD, as well as a nuclear family. The authors detect a total of 105 potential deleterious variations, which are enriched in ear or heart development-related genes. The paper also discusses genes and functions potentially affected by these deleterious variations. The study suggests that a combination of genetic variations, including SNVs and CNVs, may play a crucial role in the comorbidity of microtia and CHD. The paper provides new insight into the underlying mechanisms for the comorbidity of microtia and CHD and highlights the importance of genetic analysis in understanding the genetic basis of complex disorders. Overall, the study is well-structured and provides clear and concise methods, making it a valuable contribution to the field of microtia and CHD genetics.

Point 1: The paper presents a clear and concise description of the methods and results for SNVs and CNVs reports. The discussion section may benefit from a minor revision, as the first paragraph appears somewhat repetitive when compared to the first paragraph of the introduction. 

Response 1: Thank you for your suggestion. We have carefully revised the discussion section, especially removing redundant content that was previously included in both the first paragraph of the introduction and discussion section (lines 34-38, 42-43, 246-249,250-253). In addition, we have checked the spelling of the entire paper carefully to improve the readability and clarity of the manuscript.

Reviewer 3 Report

The paper is interesting.

Line 47: check “most frequent SVs found” à SNVs?

Line 87: check “defined phred quality threshold” à Phred?

Line 97: replace “single nucleotide variation (SNV)” to “SNV”

Table 2: place the genes in italics.

Line 286, 293: check font and size.

Line 314, 315: review typing

Author Response

The paper is interesting.

Point 1: Line 47: check “most frequent SVs found” à SNVs?

Response 1: Thank you for your suggestion. Here SV is the abbreviation of structural variation, and the corresponding abbreviation has been marked in the first appearance on line 44.

Point 2: Line 87: check “defined phred quality threshold” à Phred?

Response 2: Thank you for your suggestion. We have changed “defined phred quality threshold” to “defined Phred quality score” (line 98).

Point 3: Line 97: replace “single nucleotide variation (SNV)” to “SNV”

Response 3: Thanks for your suggestion, the error has been corrected (line 108).

Point 4: Table 2: place the genes in italics.

Response 4: The genes in Table 2 have been changed to italics (lines 225-226).

Point 5: Line 286, 293: check font and size.

Response 5: Thanks for your suggestion, we have revised the font and size (lines 292, 296, 302).

Point 6: Line 314, 315: review typing

Response 6: We have checked all the typing carefully.

Reviewer 4 Report

Well designed and well written study. It would be good if you provide a more clear conclusion supporting your hypothesis. 

Author Response

Point 1: Well designed and well written study. It would be good if you provide a more clear conclusion supporting your hypothesis.

Response 1: Thank you for your suggestion. Based on the analysis results (summarized in Table S5, please see the attachment), we further refined the conclusion to better support the hypothesis. Given the high heterogeneity of microtia and CHD, the comorbidity of these two diseases is unlikely to be caused by a few specific loci but rather may be the result of the combined effects of multiple variations in multiple genes. In our study, deleterious variants were significantly enriched in serval genes (SUSD2, TBX1, DGCR8, CLTCL1, TANGO2 and UFD1) and the microduplication fragment that may present genetic factors relevant to the etiology. The candidate variants and genes deserve further exploration to understand better the specific mechanisms of microtia and CHD comorbidity(lines 339-345). In addition, we have checked the spelling of the entire paper carefully to improve the readability and clarity of the manuscript. 

Reviewer 5 Report

Genetic screening for CHD in patients with microtia

1.      What is the objective of paper? The lines 57-61 explain de main objective but it is not clear, why the authors explore only 22q11 region? In the era of exome why not explore all genes? It is a big limitation

2.      Line 47 define the acronym SV.

3.      The association between heart disease and other genetic conditions such as CHARGE, Goldenhar.. has not been described in introduction and discussion. The title of paper is general about genetic screening and the results and discussion is focused on 22q11 region.

4.      Methods: is a custom gene panel of NGS or clinical exome or whole exome, parents were analysed in a trio approach or Sanger? For CNV author explore all genome becaude adupliations are detected in cgromosomes 1, 2, …. (Figure 4)

5.      The definition of types of microtia (1rst degree,…) described in Table 1 could be described in section 2.

6.      In the result section:

a.       there many technical results about filtering parameters, quality data, depth .. may be reduced.

b.      How many cases with CHD and microtia have a pathogenic variant? (2 cases with deleterious variants en TBX1) (line 264)

c.      Table 2 included variant in control and cases? Some of them are frequent in control population (SCARF2, TBX1, CLTCL1…) so their pathogenicity is debatable. Other genes are recessive (SCARF2, TANGO2)

d.      there is little description/discussion about variants in RIMBP3, DGCR8, PI4KA, and UFD1 where there is little medical literature in pubmed

e.      The microduplication regions detected are pathogenic?

7.      In the discussion section:

Should be redo because the discussion has to be according to the results.

Author Response

Point 1: What is the objective of paper? The lines 57-61 explain de main objective but it is not clear, why the authors explore only the 22q11 region? In the era of exome why not explore all genes? It is a big limitation

Response 1: Thank you for your valuable suggestions. The main purpose of our study is to explore the genetic mechanism underlying the comorbidity of microtia and congenital heart disease. We have revised the introduction section to make it clearer (lines 60-63). Initially, we noticed that some patients with microtia also had congenital heart disease in clinical practice. Through literature review, we found that microtia and congenital heart disease often coexist in the phenotypes of patients with 22q11.2-related syndromes. Then, we hypothesized that the co-morbidity of the two diseases is most likely associated with variants in this region. Thus, we performed an in-depth sequencing of the 22q11.2 region (mean depth = 80×) and identified genes rich in deleterious variants, such as SUSD2, TBX1, DGCR8, CLTCL1, TANGO2 and UFD1, which may explain the comorbidity in some patients (lines 339-345). Due to budget constraints, we were unable to initiate a whole-genome study at the beginning of this research, which is certainly regrettable. In the subsequent series of studies, we will continue to enhance and refine our research.

Point 2: Line 47 define the acronym SV.

Response 2: Thank you for your suggestion. The acronym SV was defined in the first appearance on line 41.

Point 3: The association between heart disease and other genetic conditions such as CHARGE, and Goldenhar.. has not been described in the introduction and discussion. The title of the paper is general about genetic screening and the results and discussion are focused on the 22q11 region.

Response 3: Thank you for your advice. We have added relevant content in the introduction and discussion section, hoping to better explain the topic (lines 34-38, 258-263). Considering that this paper focuses on the 22q11.2 region, we have made an appropriate modification to the title to better reflect the theme. The new title is “Genetic screening of targeted region on the chromosome 22q11.2 in patients with microtia and congenital heart defect” (lines 2-3).

Point 4: Methods: is a custom gene panel of NGS or clinical exome or whole exome, parents were analysed in a trio approach or Sanger? For CNV author explore all genome becaude adupliations are detected in cgromosomes 1, 2, …. (Figure 4)

Response 4: Thank you for your suggestion. We used custom-targeted sequence capture probes to capture the 22q11.2 region (8M, including coding and non-coding regions) for all samples, including 19 sporadic samples and the trio family. We fine-tuned the corresponding content in the methods section to make the description more accurate (lines 89-90). CNV detection was only focused on this region. In Figure 4, DUP and DEL are used to represent duplication and deletion, respectively, both located on chromosome 22 and not involving other chromosomes.

Point 5: The definition of types of microtia (1rst degree,…) described in Table 1 could be described in section 2.

Response 5: Thanks for your suggestion. The content regarding the definition of microtia grading has been moved to section 2.1 (lines 75-80).

Point 6: In the result section:

a.there many technical results about filtering parameters, quality data, depth .. may be reduced.

b.How many cases with CHD and microtia have a pathogenic variant? (2 cases with deleterious variants en TBX1) (line 264)

c.Table 2 included variant in control and cases? Some of them are frequent in control population (SCARF2, TBX1, CLTCL1…) so their pathogenicity is debatable. Other genes are recessive (SCARF2, TANGO2)

d.there is little description/discussion about variants in RIMBP3, DGCR8, PI4KA, and UFD1 where there is little medical literature in pubmed

e.The microduplication regions detected are pathogenic?

Response 6: Based on a series of valuable suggestions, we have made the following revisions in turn.

  1. We have moderately reduced the technical results to make the emphasis on certain parts of the results clearer (lines 172-174, 185-189).
  2. There are 105 potential deleterious variations, and each patient carries at least one deleterious variation, especially three individuals who carry variants of known susceptible genes for microtia or CHD (lines 270-272).
  3. Yes, table 2 showed the overall status of mutations in the control and case groups, with 103 and 19 samples, respectively. Although some variants appear to occur more frequently in the control group, considering the difference in sample size, the burden of these mutations is significantly higher in the case group than in the control group (P < 0.05). This suggests that these genes may be associated with the disease in the case group.
  4. Yes, there are relatively few research papers on the functions of these genes. We performed a literature search and supplemented relevant content in the discussion section (lines 310-327).

DGCR8 encodes an essential component of the microprocessor complex that mediated the biogenesis of microRNA[1]. The abnormal microRNA biogenesis due to DGCR8 deficiency contributes to structural abnormalities in the heart and vasculature and neuronal deficits [2, 3]. In mice, loss-of-function of DGCR8 in NCCs results in congenital heart abnormalities that are characteristic of 22q11.2DS, and heterozygous loss-of-function mutations of DGCR8 result in neuronal deficits [2, 4].

RIMBP3 is mostly expressed outside of the brain. As a member of the Rab3-interacting molecules (RIM) binding proteins (RIMBPs), RIMBP3, together with RIMBP1 and RIMBP2, regulate the fidelity of synaptic transmission at central synapses and play a potential role in regulating the fusion and release of neuronal vesicles by stabilizing the active zone structure in the presynaptic region and locating presynaptic ion channels [5, 6].

Previous studies suggested that PI4KA is a candidate gene for schizophrenia susceptibility and plays an important role in early brain development[7, 8]. Ubiquitin Fusion-Degradation Protein-1 Like (Ufd1l) encoded by the gene UFD1 serves as a ubiquitin recognition factor linking the ubiquitination and degradation process of proteins. p97 in mammals associated with the cofactor Ufd1-Npl4 form a functional conservative complex, which can mediate the transport of endoplasmic reticulum proteins into the cytosol for degradation[9-11]. As a housekeeping gene, UFD1 is mainly expressed in pharyngeal arches, palatal precursors, and ears, heart, brain during development, and the haploinsufficiency of UFD1 is supposed to be responsible for the pathogenesis of 22q11.2 DS [12-14]. A previous study reported the association between the bicuspid aortic valve and reduced expression of UFD1, further supporting a role of UFD1 in normal heart formation [15].”

e. Through CNV analysis, we found that nearly 60% of the samples (11 out of 19) carried a duplicated fragment (DUP3), which was validated in an independent cohort of 160 samples. This suggests that the fragment may be associated with disease onset. The fragment is located on the SUSD2 gene, which plays a dual role in different stages of neural development. SUSD2 can alter axon and dendritic outgrowth in young neurons, and regulate excitatory synapse numbers at a later stage. Despite the absence of research linking SUSD2 to the development of the ear and heart directly, this still provides a lead for future functional study.

Point 7: In the discussion section: Should be redo because the discussion has to be according to the results.

Response 7: Thanks for your suggestion. We have reorganized the discussion around the main results of the paper (summarized in Table S5). We hope that the revised content will satisfy you (lines 250-253,258-265, 310-327, 339-343).

references

  1. Wang Y.;Medvid R.;Melton C.;Jaenisch R.;Blelloch R. DGCR8 is essential for microRNA biogenesis and silencing of embryonic stem cell self-renewal. Nat. Genet. 2007,39,380.
  2. Stark K. L.;Xu B.;Bagchi A.;Lai W.-S.;Liu H.;Hsu R.;Wan X.;Pavlidis P.;Mills A. A.;Karayiorgou M. Altered brain microRNA biogenesis contributes to phenotypic deficits in a 22q11-deletion mouse model. Nat. Genet. 2008,40,751.
  3. Smith T.;Rajakaruna C.;Caputo M.;Emanueli C. MicroRNAs in congenital heart disease. Annals of translational medicine. 2015,3.
  4. Chapnik E.;Sasson V.;Blelloch R.;Hornstein E. Dgcr8 controls neural crest cells survival in cardiovascular development. Dev. Biol. 2012,362,50-56.
  5. Acuna C.;Liu X.;Gonzalez A.;Südhof Thomas C. RIM-BPs Mediate Tight Coupling of Action Potentials to Ca2+-Triggered Neurotransmitter Release. Neuron. 2015,87,1234-1247.
  6. Gao T.;Zhang Z.;Yang Y.;Zhang H.;Li N.;Liu B. Impact of RIM-BPs in neuronal vesicles release. Brain Res. Bull. 2021,170,129-136.
  7. Jungerius B. J.;Hoogendoorn M. L. C.;Bakker S. C.;van't Slot R.;Bardoel A. F.;Ophoff R. A.;Wijmenga C.;Kahn R. S.;Sinke R. J. An association screen of myelin-related genes implicates the chromosome 22q11 PIK4CA gene in schizophrenia. Mol. Psychiatry. 2008,13,1060-1068.
  8. Pagnamenta A. T.;Howard M. F.;Wisniewski E.;Popitsch N.;Knight S. J.;Keays D. A.;Quaghebeur G.;Cox H.;Cox P.;Balla T.;Taylor J. C.;Kini U. Germline recessive mutations in PI4KA are associated with perisylvian polymicrogyria, cerebellar hypoplasia and arthrogryposis. Hum Mol Genet. 2015,24,3732-3741.
  9. Ye Y.;Meyer H. H.;Rapoport T. A. Function of the p97–Ufd1–Npl4 complex in retrotranslocation from the ER to the cytosol: dual recognition of nonubiquitinated polypeptide segments and polyubiquitin chains. The Journal of cell biology. 2003,162,71-84.
  10. Ye Y.;Meyer H. H.;Rapoport T. A. The AAA ATPase Cdc48/p97 and its partners transport proteins from the ER into the cytosol. Nature. 2001,414,652-656.
  11. Bays N. W.;Hampton R. Y. Cdc48–Ufd1–Npl4: stuck in the middle with Ub. Curr. Biol. 2002,12,R366-R371.
  12. Novelli G.;Mari A.;Amati F.;Colosimo A.;Sangiuolo F.;Bengala M.;Conti E.;Ratti A.;Bordoni R.;Pizzuti A. Structure and expression of the human ubiquitin fusion–degradation gene (UFD1L). Biochimica et Biophysica Acta (BBA)-Gene Structure and Expression. 1998,1396,158-162.
  13. Yamagishi;H. A Molecular Pathway Revealing a Genetic Basis for Human Cardiac and Craniofacial Defects. Science.283,1158-1161.
  14. Rizzu P.;Lindsay E. A.;Taylor C.;O’Donnell H.;Levy A.;Scambler P.;Baldini A. Cloning and comparative mapping of a gene from the commonly deleted region of DiGeorge and Velocardiofacial syndromes conserved inC. elegans.7,639-643.
  15. Mohamed S. A.;Hanke T.;Schlueter C.;Bullerdiek J.;Sievers H.-H. Ubiquitin fusion degradation 1–like gene dysregulation in bicuspid aortic valve. The Journal of Thoracic and Cardiovascular Surgery. 2005,130,1531-1536.

Round 2

Reviewer 5 Report

no comments